# Artificial Intelligence-Based Technological-Oriented Knowledge Management, Innovation, and E-Service Delivery in Smart Cities: Moderating Role of E-Governance

Syed Asad Abbas Bokhari [1]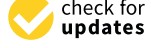 and Seunghwan Myeong [2,*]

1  The Center of Security Convergence & eGovernance, Inha University, Nam-gu, Incheon 22212, Korea
2  Department of Public Administration, Inha University, Incheon 22212, Korea
*  Correspondence: shmyeong@inha.ac.kr; Tel.: +82-1057071108

**Abstract:** The fundamental goal of this research is to investigate the quantitative relationship between technology-oriented knowledge management, innovation, e-governance, and smart city performance using knowledge management-based service science theory and diffusion of innovation theory. Previous research has found a connection between knowledge management, innovation, e-governance, and e-service delivery. We believe these are not only direct connections but also contextual and interactive relationships, so we explored the significance of innovation as a mediator between knowledge management and e-service delivery. Furthermore, we investigated the moderating impact of e-governance on the relationship between innovation and e-service delivery. A survey questionnaire was administered to the population of public officers, entrepreneurs, and citizens, from metropolitan cities for data sampling, and SPSS was applied to analyze data of 569 participants collected from South Korea, Pakistan, Japan, and Bangladesh. We discovered from the analysis that the direct relationships are contextual because innovation mediates the relationship between knowledge management and e-service delivery, and e-governance plays a moderating role in the relationship between innovation and e-service delivery. Based on the outcomes from quantitative analysis, all our proposed hypotheses in this study were supported significantly.

**Keywords:** technology-oriented knowledge management; innovation; e-governance; e-service delivery; smart city performance

## 1. Introduction

The recognition of "resources" or "capabilities" that permit organizations to identify, generate, convert, and disseminate knowledge is critical to realizing the successes and failures of knowledge management (KM) within corporations. The structural, technical, and cultural elements that enable KM's intensification of social capital are termed KM infrastructure [1,2]. The innovation facet is related to the technologically enabled affiliations that emerge within organizations [3], and organizations can ambitiously be organized by a 'smart city' [4]. The presence of norm and trust mechanisms, as well as collaborative learning environments, is signified by the institutional and cultural dimensions.

The appraisal of the KM infrastructure that allows the institutions to identify, develop, transform, and disseminate knowledge is crucial in understanding the strengths and weaknesses of KM initiatives and their impact on different elements.

Numerous scholars have stressed the significance of knowledge systems and applications in knowledge management [5,6]. Previous KM research has been segmented in that it has described a few components of KM performance but has not offered a comprehensive viewpoint of KM impact on other organization attributes such as innovation performance and smart city performance. Most researchers have examined the association between KM enablers, procedures, or outcomes in exclusion. For instance, Gold et al. [7] proposed that the infrastructure of knowledge (culture, technology, structure) and the

process of knowledge (attainment, adaptation, submission, and security) have a direct influence on organizational effectiveness. However, they ignored the correlation between knowledge management and innovation. While Lee et al. [8] demonstrated the cooperative relationships between knowledge management enablers, knowledge creation procedures, knowledge management transitional outcomes, and organizational performance, their research did not contemplate the entire knowledge process and its direct and indirect impact on performance.

Currently, the emphasis on innovation- and technology-led evolution is on innovation hubs and inventive centers, smart technological localities, and Living Laboratories that test innovative products [9]. KM has taken power from the confines of the corporate world and enlarged into other socio-economic fields such as education and governance [10]. Major global institutions, including the UN, the World Bank, the EU, and the Organization for Economic Cooperation and Development (OECD), have incorporated knowledge management frameworks into their domestic and global strategic planning. It has become obvious that there is a significant association between knowledge management and urban development, as city activities can be deliberately created to enable knowledge cultivation. Many scholars are looking into 'knowledge cities' [11] and knowledge-based smart city development [12]. Integrated knowledge and innovation are crucial determinants of the smart city's rhetoric and execution. Recent technology capabilities would never have the same impact on smart cities if they had never been entrenched in knowledge and innovation [13]. The extensive knowledge market was essential for the implementation of the paradigm of cities; it is one of two intellectual components that comprise the contemporary concepts about a smart city, its implementation, and enhanced performance.

The term "smart city" is frequently linked with the notion of a digital city, with the extensive use of technology, especially its performance in governance, surveillance, mobility, education, health, and telecom infrastructure [14]. Nevertheless, the idea of a smart city extends beyond technology to include other predictors of innovation and governance, such as technological innovation, institutional innovation, social innovation, e-governance, e-government, and smart governance issues [15,16]. Considering the importance of city governance and administration, as well as collaboration between different stakeholders, to meet the optimum city performance, innovation, expansion, sustainable development, and liveability [17], we aim to investigate how smart governance affects smart city performance directly and also moderates the association between innovation and city performance.

The main objective of our study is to examine the relationship between technology-oriented knowledge management, innovation, e-governance, and smart city performance with the help of knowledge management that is based on service science theory [18] and diffusion of innovation theory [19], as service science theory discusses the use of knowledge that is collected through citizen and artificial intelligence can help to improve and optimize city's service delivery. Diffusion of innovation theory refers to the procedure by which people espouse a new concept, product, practice, and ethos. Further, we will investigate the indirect mediating role of innovation in the relationship between technology-oriented KM and smart city performance and the indirect moderating role of e-governance in the relationship between innovation and smart city performance. Previous scholars examined the direct impact of knowledge management on innovation [20,21], the impact of innovation on smart city performance [22], and the effect of e-governance on smart city performance [23,24], but only a few explored the indirect relationship between these constructs [25]. Our study will contribute to the existing literature by investigating indirect associations to know-how innovation that mediates the relationship between the integration KM and city performance and how e-governance strengthens the relationship between innovation and smart city performance.

The subsequent section briefly outlines the literature on KM, innovation, e-governance, and smart city performance. Next, Section 3 describes the research method that was used to find relevant outcomes for this study. Finally, Section 4 describes the research findings, while Section 5 discusses the recommended next steps for smart city research from the

perspective of knowledge management. Section 6 concludes with policy implications and a conclusion.

## 2. Background and Hypotheses

### 2.1. Technological-Oriented KM, Innovation, and Smart City Performance

For numerous eras, the practice of knowledge management (KM) has attracted the attention of researchers and experts alike. Academics and professionals have focused their efforts on the discussion of how to effectively employ KM in contemporary organizations to achieve better outcomes [6,8]. The fundamental KM approach and its application to accomplish benefits of performance and competitive advantages are critical success factors in this context [26,27]. Considering the significance of technological innovation and knowledge sharing in our economic system, knowledge management will hold a significant role in the corporate in the future [28]. Consequently, the digital revolution and the increased high-tech innovations in various disciplines will absorb tedious tasks, abandoning only complicated operations for highly competent, primarily white-collar workers [29]. Concurrently, new forms of knowledge are developed because of such new technologies, leading to new prerequisites for administering knowledge [30]. Previous literature suggests one of the main approaches of KM, which is referred to as technology-oriented KM, and it follows a codified strategy to find explicit knowledge that is stashed in external databases [31]. Digitization can effectively process enormous amounts of heterogeneous data, knowledge, and information by employing AI and associated technologies. There are two aspects that distinguish AI applications, which determine our understanding of knowledge and how it is managed in institutions. First, AI algorithms can process data and discover trends autonomously, perhaps more effectively than people. As a corollary, these evolutionary computations can instantly develop important types of knowledge from data [30].

Smart city governments are constantly under pressure to enhance public service delivery with a citizen-friendly approach to digital transformation. Local governments in smart cities are constantly interested in improving the citizen-friendly delivery of public services in the age of technology revolution to enhance efficiency. Instead of focusing on a specific range of services for target markets, as is common in the private sector, municipal government services must manage a broad, diverse array of services that must be delivered to all inhabitants [32]. Even though distinctive clusters of residents will have unique attributes and expectations, access to public services and information must be guaranteed [33], while the cost efficiency of service delivery must be sustained.

Knowledge sharing is critical to the principle of Knowledge Alignment because knowledge integration cannot be easily accomplished without sharing. Consequently, numerous previous researchers found no association between Knowledge Stock and Knowledge Integration [34], which is not surprising given that the level of expertise does not indicate proclivity to share. It is consistent with prior research, which discovered that knowledge had little or no direct impact on performance [35]. Subject Matter Experts may be reluctant to share their knowledge with non-domain professionals for various reasons, including power, language differences, and time constraints [36]. On the contrary, most organizations claim that an effective and efficient KM process will benefit organizational performance. As a result, knowledge management is widely accepted as an important predictor of organizational innovation or performance [37]. However, there are some differences in the outcomes of KM sub-processes or sub-dimensions and organizational performance.

Performance is a common thread in most disciplines, such as social science and management, and it is significant to academics and practitioners. Although the relevance of the notion of performance is broadly accepted, the intervention of performance in study designs is perhaps one of the most difficult issues that is encountered by academic researchers today. With the quantity of literature on the subject constantly growing, there appears to be little hope of achieving alliance on basic terminology and interpretations. Some people have expressed their dissatisfaction with this concept. Consequently, smart city performance should be included in electronic service delivery by smart cities in this

study [38]. From a traditional standpoint, organizational performance is usually associated with economic performance [39], and the financial benefits of organizational effectiveness are strongly tied to the company's performance [40]. Darroch's [37] analysis employs contrasting and individually introspective performance indicators, such as "Our company is more profitable than the industry average," and individual introspective performance indicators, such as "We are more profitable than we were five years ago." These performance indicators include both financial and non-financial indicators.

Nevertheless, similar to any other organizational resource, effective technology-oriented knowledge management through artificial intelligence should contribute to key attributes of smart city performance, such as e-service delivery [41].

Furthermore, as smart cities improve their AI-based knowledge management, they can achieve optimal e-services solutions to satisfy the needs of their citizens [30]. Smart cities can acquire and use knowledge more productively with increased AI-based knowledge management capabilities, resulting in above-average performance. Thus, we propose:

**Hypothesis 1.** *Higher the AI-based technology-oriented knowledge management, the higher the likelihood that a smart city offers e-service delivery to citizens.*

When considering the association between Knowledge Management and innovativeness, we first begin with Schumpeter. According to him, integrating established theoretical and physiological ingredients is known as an innovation [42]. Specifically, innovation is the process of combining an organization's existing knowledge capital to generate new knowledge. Consequently, an innovative business's ultimate focus is reorganizing current knowledge assets while researching new knowledge [43]. Knowledge exploration and manipulation have been proven to contribute to the innovativeness of an organization and its performance [44]. Numerous studies on the significance of Knowledge Management in innovation has been undertaken. The outcomes of Du Plessis [21] supported the crucial importance of knowledge management in knowledge processing capability and hence in the incidence and interactivity of innovation. Huergo [45] presents statistical evidence supporting the positive effect of technology management on an organizations' innovation success. Brockman and Morgan [46] argue that KM techniques such as "innovative information use," "efficient information gathering," and "shared interpretation" improve the efficiency and innovativeness of new products. Theoretical approaches provide vague arguments about a particular emphasis on "demand-driven" or "collaborative" knowledge management techniques. Incredibly strong relations in a knowledge-sharing community may constrain the innovation process due to redundancy [47]. On the other hand, a shared knowledge base enhances intellectual capital within the society [48].

Knowledge management systems, particularly ICT elements, emerge to enhance the efficiency and at least perceived progress [49]. It is compatible with the outcomes of knowledge management in businesses, which unearth statistical evidence proving enterprises with superior knowledge management employ their resources effectively, increasing innovation [21]. The findings of previous case studies offer conflicting results too. Darroch et al.'s findings are an excellent illustration. Darroch [37] discovered no substantial advantages. A further component of the KM-innovation connection is how knowledge management influences distinctive forms of innovation. According to Darroch and McNaughton [50], different kinds of innovation demand different resources and a unique knowledge management strategy, such as technology-oriented knowledge management. They examined the impact of knowledge management on three different kinds of innovation. As per their observations, diverse KM initiatives are significant for different innovations. Consequently, we believe that different knowledge management will influence different aspects of innovation success, as well as the velocity, reliability, and magnitude of innovation success. Hence, we propose:

**Hypothesis 2.** *Higher the AI-based technology-oriented knowledge management, the higher the likelihood that a smart city will have more innovation success.*

Innovation is a modern concept, discipline, or artifact that a person or entity perceives as novel. When an innovation emanates, diffusion occurs, which implies interacting or distributing the innovation reports to the intended group [51]. According to the theory of diffusion of innovations, diffusion of innovation emerges when potential consumers become informed of the innovation, analyze its significance, and decide, based on their assessment, to incorporate or reject the innovation and demand evidence of the deployment or disapproval decision [51]. These mechanisms eventually occur through a platform among citizens (consumers). Diffusion of innovation considers individual and societal elements that influence an adoption decision or abandon a particular innovation. Rogers contends that cognitive and social factors, as well as environmental and contextual aspects, may influence the diffusion of innovation.

Service innovation, defined as "new developments in service processes involved in delivering core products and services" [52], can be defined as a group of enhanced efficiency for delivering existing services or products [53]. E-service innovation focuses on services that are provided mostly through digitized network connectivity, demonstrating the types of companies that employ internet technologies to optimize service delivery and adapt the services that suit the client's demands. E-service innovation improves value by facilitating service providers to leverage digital strategies for improving customer–healthy relationships and reducing service output uncertainty [54]. External data can be consolidated with digital knowledge acquired through the internet and other useful information to maximize the effectiveness of service delivery [55]. E-service innovation can be investigated by identifying the qualities that distinguish it from all other innovations for improved service delivery [56]. Consequently, e-service innovations can encourage organizations to provide enhanced customer value while improving e-service delivery.

Another relationship that is investigated in this study is the link between innovation and smart city performance, which is a city's capacity to provide e-service delivery. Previous research established a significant positive association between innovation and performance [37,57,58]. Hence, we proposed the following hypotheses on this basis:

**Hypothesis 3.** *Higher the innovation, the higher the likelihood that a smart city will provide e-service delivery.*

**Hypothesis 4.** *Innovation mediates the relationship between AI-based technology-oriented knowledge management and e-service delivery.*

### 2.2. Moderating Role of E-Governance

*E-governance* is defined as "the public sector's use of information and communication technologies to improve information and service delivery, encouraging citizen participation in the decision-making process, and making government more accountable, transparent and effective . . . its objective is to engage, enable and empower the citizen" [59]. Citizens, corporations, governments, and institutions all benefit from e-governance. Citizens get benefits from electronic services that are affordable, convenient, instantaneous, efficient, transparent, and equitable around the clock; businesses take advantage of lower time in registration of new business set-up, get assistance in undertaking e-commerce business, superior compliance to regulatory standards to conduct business, convenient and more transparent while doing business with government through e-tendering, and preventing corruption during finance clearing from government compensation by employing e-banking. Government institutions benefit from up-to-date information for proper policy decisions and regulatory control; quick handling of provided data for improved decision-making; efficient management; stronger propagation of regulatory norms; improved results in regulatory mechanisms such as taxation; higher performance in social sectors such as

health, education, and social welfare; and developing a positive impression of dynamic modern government in public.

Smart city governments constantly look for modern techniques to provide the quality of public services. E-Government is one indication of a drastic transformation in service delivery to citizens, in which unique information and communication technologies (ICT), mechanisms, organizational structures, and management systems are launched to promote public significance and generate positive change in people's lives [60]. During this evolution, a significant number of innovations were implemented. Compared to the corporate sector, where organizations attempt to maximize competitiveness to generate profit, government institutions strive to innovate to generate better performance. Further, public sector services are poised to generate public performance and improve desired public outcomes. The three main principles of public sector innovations are novelty, execution, and implications, which lead to better public outcomes such as reliability, performance, transparency, and user satisfaction [61].

Service delivery innovation is among the best-acclaimed innovations in public sector organizations in Eu countries; according to the 2010 European Union's Yardstick, 66 percent of organizations across the EU-27 report experienced incorporated innovations in public services [62,63]. System and governance strategies for innovation have been identified as the most prevalent, particularly at the domestic level. Environmental challenges, increasing population, and poverty have highlighted the use of creative and innovative approaches to the challenges confronting public services in European cities. As novel approaches to address the most complex urban challenges, modern e-governance frameworks, organizational techniques, and transparency have been proposed [64]. Technology innovation has recently boosted governments' capabilities to perform the necessary methodologies and procedures to achieve this [65].

ICT has been invented to provide an intensifying range of services, provide people access to online platforms, and mitigate service delivery costs. These activities fall under the umbrella of e-government, which aims to "enable and improve the efficiency with which government services and information are provided to citizens, employees, businesses, and government agencies" [66]. In terms of communication channels for the delivery of government services, the online channel is likely to be the top priority for governments, owing to its cost-effectiveness [67]. As a result, governments are interested in their citizens' adoption of the online service delivery channel. Consequently, the essence of government portals must concentrate on those unique requirements and strive to satisfy "consumers" (inhabitants, citizens, and enterprises) [68]. Considering these requirements, governments must choose an online service delivery model that integrates structure and content to improve performance. Hence, we propose our hypotheses as follows:

**Hypothesis 5.** *Higher the implementation of e-governance in a smart city, the higher the likelihood that a smart city offers e-service delivery to citizens.*

Several previous studies have utilized governance as moderating variable to investigate their constructs, for example, moderating the role of governance mechanisms on the relationship between ESG disclosure and firm performance [69]; moderating the role of governance on the relationship between free cash flow and earning management [70]; moderating role of governance heterogeneity on the relationship between psychological ownership, knowledge sharing, and entrepreneurial orientation [71]; and moderating role of governance environment on the relationship between risk allocation and private investment [72]. We assume that e-governance is best suited to be applied as a moderating variable to investigate the relationship between innovation and smart city performance. Hence, we propose:

**Hypothesis 6.** *Relationship between innovation and smart city performance is strengthened with the moderating impact of e-governance.*

## 3. Research Methodology

### 3.1. Sampling

Increasingly, researchers are combining mixed-method approaches to establish a deeper level explanation for this phenomenon that is under investigation, improve the validity of the results, and explain conflicting outcomes [53]. This study used a quantitative survey technique to collect data for testing the proposed research model and hypotheses. The quantitative survey was carried out from January 2022 to May 2022. Following that, interviews were performed. We interviewed public officers in target cities in Pakistan in April 2022 to help interpret and understand the statistical results, thereby strengthening the outcomes. The data were acquired from a sample of South Korea, Pakistan, Bangladesh, and Japan public officials and citizens that were directly or indirectly involved in public service delivery decision-making. This assessment threshold was developed on the assumption that senior officials and citizens would necessitate the presence of some system to ensure knowledge management. The most qualified individuals in each department were identified and requested to respond to the survey, presuming that they would be qualified to comment on the transmission of knowledge throughout the organization instead of one or two departments.

The survey's administration took place in three stages. After identifying the population of public officers, entrepreneurs, and citizens, from metropolitan cities with a population of 600,000 or more in South Korea, Pakistan, Japan, and Bangladesh, a pre-notification mail describing the objective of the study and proclaiming the impending influx of the survey was sent to targeted respondents. The justification for choosing these four countries was that South Korea and Japan are East Asian developed economies with strong e-governance and e-services for their citizens [73]. In contrast, Pakistan and Bangladesh are South Asian emerging economies striving to design and implement such governance and services [74], so it is essential to evaluate respondents' perceptions from different geographic areas from the same continent. According to our best knowledge and observation, only a few studies have yet been undertaken in the comparative sense of such Asian regions [75].

Two weeks later, a set of questionnaires was forwarded to the targeted respondents, including shared online on different social media websites. The effective usable sample size was 569. Although very few experimental investigations on knowledge management were identified in the existing literature, it is hard to determine how age, education, experience, or nationality may have influenced the findings. To test for quasi-bias, a spontaneous cross-section of 90 participants who had not responded was chosen and delivered a short survey questionnaire to fill. The brief questionnaire was completed by 24 (26.7 percent) of this group. ANOVA analyses reported no difference in the mean replies from early, late, or non-respondents and thus no substantial variation between each segment of the respondents. Table 1 describes the respondents' age, education, experience, and nationality characteristics.

**Table 1.** Personal characteristics of the survey respondents.

| Characteristic | Category | N | % |
|---|---|---|---|
| Age | 18 to 30 years | 297 | 52 |
| | 31 to 40 years | 188 | 33 |
| | 41 to 50 years | 58 | 10 |
| | More than 50 years | 26 | 05 |
| Education | PhD degree | 55 | 10 |
| | Master's degree | 174 | 30 |
| | Bachelor's degree | 340 | 60 |
| Experience | 1 to 10 years | 176 | 31 |
| | 11 to 20 years | 326 | 57 |
| | 21 to 30 years | 60 | 11 |
| | More than 30 years | 7 | 01 |
| Nationality | South Korea | 380 | 67 |
| | Japan | 31 | 05 |
| | Pakistan | 109 | 19 |
| | Bangladesh | 49 | 09 |

*3.2. Construct Measurement*

A survey questionnaire was constructed to evaluate the four possible phenomena that were under study: (a) technology-oriented knowledge management (KM); (b) innovation; (c) e-governance; and (d) smart city performance. All of the variables were assessed with components that had previously been substantiated in research. The survey questionnaire items were paraphrased to address the perspective of this study explicitly.

### 3.2.1. Knowledge Management

Knowledge management was adapted from [76], which designed three scales to evaluate KM behaviors and practices: acquiring, disseminating, and responding to knowledge. There were eight factors that captured those:

- processes for acquiring knowledge about traffic violations through the database (KM1)
- processes for acquiring knowledge about our citizens' behavior through AI (KM2)
- process for acquiring knowledge about new services (KM3)
- process for acquiring knowledge about competitors within our private industry (KM4)
- feedback from projects through the database to improve subsequent projects (KM5)
- processes for exchanging knowledge with our private business partners (KM6)
- process for benchmarking performance through the database (KM7)
- teams that are devoted to identifying best practices for services (KM8)

### 3.2.2. Innovation

This paper employs the adapted [77] typology of Innovation. In this context, *Innovation* is defined as creating groups with different areas of expertise (INN1), knowledge sharing within groups (INN2); knowledge sharing between groups (INN3); encouragement to question and reflect on the decisions (INN4); availability of physical resources to acquire new knowledge to develop new ideas (INN5); allocate time for idea generation through knowledge sharing (INN6); new or significantly improved methods of producing services (INN7); the acquisition of advanced machinery, equipment, and computer hardware for the development of new or significantly improved services (INN8); the acquisition of software for the development of new or significantly improved services (INN9); and the acquisition of existing knowledge, copyrighted works, patented and non-patented inventions, and other types of knowledge from other cities (INN10).

### 3.2.3. E-Governance

We adapted the measurement scale [78] to determine e-governance for this study. *E-governance* is defined in this context as a strategy of local government for e-government (EG1), a citizen's right to require digital communication (EG2); businesses right to require digital communication (EG3), public authority's right to require digital communication from other parts of the public sector (EG4), utilization of ICT project budget thresholds/ceilings to structure its governance processes (EG5), public services or procedures that are mandatory to use online (EG6), government priority to increase the number of mandatory online services that are aimed at citizens (EG7), government priority to increase the number of mandatory online services that are aimed at businesses (EG8), and the main national citizen portal for government services (EG9).

### 3.2.4. Smart City Performance

We utilized Eeservice delivery to measure the construct of smart city performance. The measurement scales that were used by [33] for e-service delivery were adapted to investigate this variable here. We measured e-service delivery in this perspective as the ease of enrolment of voting online for the first time in government elections (ESD1), ease of lodging personal income tax return online (ESD2), ease of renewing international passport online (ESD3), ease of renewing personal driving license online (ESD4), ease of making an official declaration of theft of personal goods to the relevant police online (ESD5), ease of obtaining a copy of a birth certificate for self electronically (ESD6), ease of obtaining a copy

of a marriage certificate for self electronically (ESD7), and ease of renewal of registration for a motor vehicle online (ESD8).

### 3.3. Analysis

The survey data were analyzed employing IBM SPSS Statistics 23 and SmartPLS 3, a multi-regression modelling approach that has gained prominence due to its precision and effectiveness. The multi-regression technique includes a regression estimation procedure, depicting quantitative and qualitative latent constructs while enforcing fundamental criteria on scale items, sample size, and redistributive assumptions. We performed an analysis in stages: (1) we evaluated the measurement model by restricting our indicators to a sequence of confirmatory factor analysis (CFA); and (2) we developed a structural model to investigate our hypotheses. SPSS 23 [79] was used for statistical analysis to substantiate the indicators and investigate the hypotheses.

To ensure that the answers were truly representative, the stimulatory effects of nonrespondent bias were mitigated by distinct participants to a sample of nonrespondents that were predicated on personal characteristics such as age, education, and experience. At the 5% level of significance, the chi-square test results found no significant difference between the three respondent groups for age ($\chi^2 = 70.323$, $p < 0.01$), education ($\chi^2 = 484.580$, $p < 0.01$), gender ($\chi^2 = 4.937$, $p < 0.01$), and experience ($\chi^2 = 423.907$, $p < 0.01$). Consequently, we asserted that this study was not concerned with nonresponse bias.

Another potential source of concern is the presence of common technique bias. By separating predictors and criterion construct objects throughout a lengthy survey question and assuring survey confidentiality, we reduced typical technique bias. The Harman one-factor test was used to look for common approach bias [80]. An unrotated confirmatory factor analysis of all the elements that were employed in this study reveals five elements with eigenvectors that were greater than one, which explains about 73% of the variation. The first (largest) component accounted for 18% of the variance. As multiple factors were collected and no single criterion accounted for more than 52% of the variation, common technique bias was not identified as a significant concern.

A convergent validity test was employed to create a measurement model of the entire self-rating scales using confirmatory factor analysis (CFA). After that, the modification index was used to select objects from the factors. The element with the highest modification index score was eliminated first, followed by the next component, until the intended goodness of fit was accomplished. Most goodness-of-fit predictors surpassed the defined cut-off criterion, but a few factor loadings were below the minimum standard of 0.5. Therefore, we excluded them from acquiring valid data for our model. The factor loadings of all factors of estimated parameters are validated to be higher than the critical value point of 0.5 [15]. We are now at the crucial stage of determining whether the conceptual framework we have defined is legitimate after it has been explained and delivered all the necessary reliability and validity tests. It was achieved by ascertaining the goodness-of-fit benchmark for the model fit. The potential to ascertain how well the model fits into the variation structure of the dataset is regarded as the goodness of fit. The CFA evaluation and research framework represent the data well based on quantitative assessment criteria. Cronbach's alpha coefficients were employed to evaluate the reliability of the metrics, and construct correlation was applied to estimate the sample's validity. The items for each variable were created using previous research. These indices have the potential to provide definitive evidence about construct reliability and validity above the threshold of 0.50.

Figure 1 illustrates our research framework, in which technology-oriented KM is depicted as an independent variable, smart city performance dependent, and innovation mediating and e-governance as a moderating variable. Our conceptual framework suggests a direct impact of technologically-oriented KM on smart city performance, which is e-service delivery, but with the integration of innovation, the direct linear relationship was transformed into a mediating relationship. Furthermore, e-governance was introduced as a moderating variable between innovation and smart city performance. Statistical mediation

and moderation analysis employ three fundamental techniques: (1) causal stages, (2) coefficient difference, and (3) coefficient product [81].

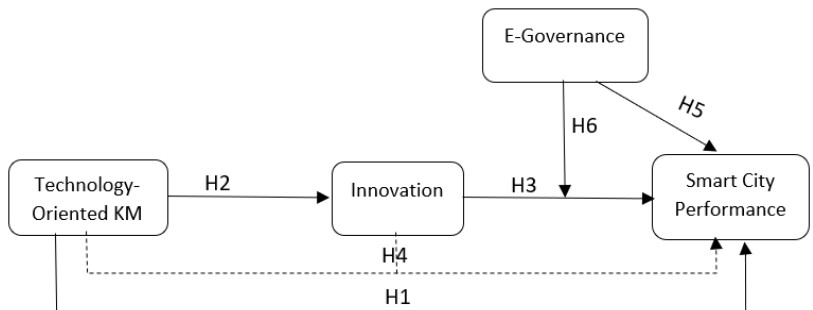

**Figure 1.** Research framework of AI-based KM on innovation and e-service delivery.

## 4. Results

Table 2 displays the item measures' standardized loading outcomes and other benchmarks, as well as the reliability and validity indicators. All of the components in the reliability analysis had factor loadings varying from 0.637 to 0.895, suggesting they were suitable for the rest of the assessment. The composite reliability indicators of all of the first-order components range from 0.903 to 0.953, which is greater than the recommended threshold of 0.70 [82]. Furthermore, the average variance that was extracted was greater than the 0.50 threshold that was suggested by [82]. The descriptive and discriminant validity of the measurements is shown in Table 3. For better discriminant validity, the square root of a construct's average variance must be greater than the square root of the construct's comparisons with the other components [83]. The findings also suggested that our components met this threshold, proving discriminant validity. An investigation of cross-loadings revealed appropriate discriminant validity as well.

**Table 2.** Construct reliability and validity Using CR, AVE, Cronbach's' Alpha, and KMO Test.

| Item | Standardized Factor Loadings | Composite Reliability | Average Variance Extracted (AVE) | Cronbach Alpha | KMO and Bartlett's Test |
|---|---|---|---|---|---|
| Cronbach Alpha = 0.971 | | | KMO & Bartlett's Test = 0.815 | | |
| KM1 | 0.742 | | | | |
| KM2 | 0.807 | | | | |
| KM3 | 0.895 | | | | |
| KM4 | 0.870 | 0.953 | 0.717 | 0.943 | 0.934 |
| KM5 | 0.852 | | | | |
| KM6 | 0.872 | | | | |
| KM7 | 0.875 | | | | |
| KM8 | 0.850 | | | | |
| INN1 | 0.717 | | | | |
| INN2 | 0.678 | | | | |
| INN3 | 0.819 | | | | |
| INN4 | 0.823 | | | | |
| INN5 | 0.817 | 0.929 | 0.569 | 0.910 | 0.928 |
| INN6 | 0.656 | | | | |
| INN7 | 0.792 | | | | |
| INN8 | 0.637 | | | | |
| INN9 | 0.809 | | | | |
| INN10 | 0.764 | | | | |

**Table 2.** *Cont.*

| Item | Standardized Factor Loadings | Composite Reliability | Average Variance Extracted (AVE) | Cronbach Alpha | KMO and Bartlett's Test |
|------|------|------|------|------|------|
| EG1 | 0.720 | | | | |
| EG2 | 0.759 | | | | |
| EG3 | 0.805 | | | | |
| EG4 | 0.743 | | | | |
| EG5 | 0.650 | 0.917 | 0.553 | 0.897 | 0.906 |
| EG6 | 0.732 | | | | |
| EG7 | 0.784 | | | | |
| EG8 | 0.735 | | | | |
| EG9 | 0.753 | | | | |
| ESD1 | 0.735 | | | | |
| ESD2 | 0.761 | | | | |
| ESD4 | 0.813 | | | | |
| ESD4 | 0.686 | 0.903 | 0.540 | 0.840 | 0.798 |
| ESD5 | 0.672 | | | | |
| ESD6 | 0.665 | | | | |
| ESD7 | 0.776 | | | | |
| ESD8 | 0.757 | | | | |

**Table 3.** Descriptive statistics, mean, standard deviation, and correlations between variables.

| | N | Mean | Std. D | Edu | Gen | Exp | KM | Inn | EGov | ESD |
|------|------|------|------|------|------|------|------|------|------|------|
| Edu | 569 | 2.293 | 1.477 | 1 | | | | | | |
| Gen | 569 | 1.453 | 0.498 | 0.073 | 1 | | | | | |
| Exp | 569 | 2.489 | 0.695 | 0.037 | 0.157 ** | 1 | | | | |
| KM | 569 | 3.522 | 0.819 | 0.109 ** | 0.049 | 0.305 ** | 1 | | | |
| Inn | 569 | 3.592 | 0.683 | 0.108 ** | 0.095 * | 0.313 ** | 0.929 ** | 1 | | |
| EGov | 569 | 3.851 | 0.602 | 0.102 * | 0.081 | 0.210 ** | 0.715 ** | 0.841 ** | 1 | |
| ESD | 569 | 3.797 | 0.625 | 0.095 * | 0.034 | 0.318 ** | 0.668 ** | 0.771 ** | 0.614 ** | 1 |

** Correlation is significant at the 0.01 level (2-tailed); * Correlation is significant at the 0.05 level (2-tailed).

We employed IBM SPSS Statistics 23 with the bootstrap technique to examine the proposed model. An evaluation of the conceptual framework, which included the coefficients of the correlation between the constructs, substantiated the hypothesized impacts and the R-square values, which suggest the proportion of the variation in the dependent constructs is expressed by their forebears. The control constructs (Model 1) were joined into the analysis model first, preceded by the main variables (Model 2), two-way interaction effect (Model 3), and moderating effects (Model 4), as suggested by [84]. Consequently, we simulated both the interactive (Models 3 and 4) and main effects on innovation (Model 2). The findings of the structural equation model analysis are demonstrated below. We concentrated on Model 3 and Model 4 because the speculated complex interactions are statistically significant.

Figure 2 illustrates the Model 4 and Model 5 paths and their significance. Technology-oriented knowledge management had a significant impact on innovation ($\beta = 0.766$, $p < 0.01$) and e-service delivery ($\beta = 0.370$, $p < 0.01$). This factor accounted for 64.9% of the variation in innovation and 65.3% of the variation in e-service delivery. Consequently, H1 and H2 are supported. H3 was supported by the fact that innovation had a significant impact on e-service delivery ($\beta = 0.935$, $p < 0.01$). The outcomes for the three control variables in the study exhibit that the respondents' gender, education, age, and experience have no impact on innovation, e-governance, or e-service delivery.

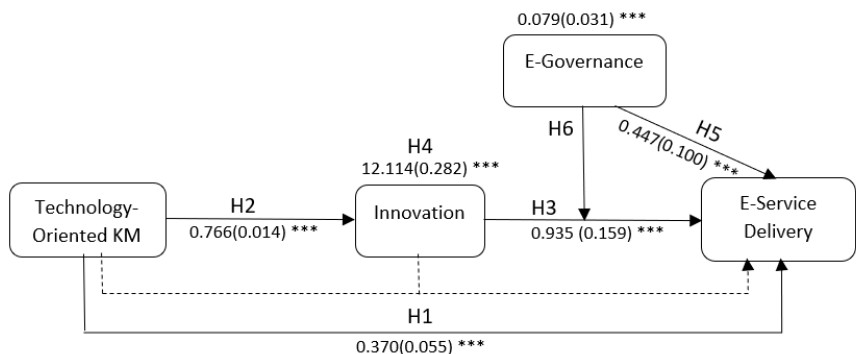

**Figure 2.** Research outcomes from multiple regression analysis. *** Impact of Technology Oriented KM, Innovation, and E-governance on E-Service Delivery.

We investigated the interaction effect of innovation on technology-oriented KM and e-service delivery and discovered a significant interactive effect ($\beta = 12.114$, $p < 0.01$). We further examined the interaction impact of e-governance between innovation and e-service delivery and found a strong significant moderating effect ($\beta = 0.447$, $p < 0.01$). This result corroborates our hypothesis that complementarity is essential in the suggested framework. Although the complementarity of internal and external dynamics may expedite synergic innovation, few investigations have been made to test this correlation. Therefore, we designed to simulate both the interactive and main impacts of innovation. When these interaction terms were included, the $R^2$ for innovation increased to 0.653. Model 5 was explained by applying multiple-regression modeling to explore the mediating role of innovation when knowledge management was a predictor variable and e-service delivery was considered an observed variable. The results in Table 5 revealed ($\beta = 12.114$, $p < 0.01$) a significant positive and indirect relationship between knowledge management and e-service delivery, hence, H4 is strongly supported.

The results of the moderating analysis are shown in Model 4 of Table 4. The findings demonstrate a direct positive relationship between e-governance and e-service delivery ($\beta = 0.447$, $p < 0.01$), strongly supporting our proposed H5. It indicated a significant and positive direct relationship between e-governance and e-service delivery. Moreover, we hypothesized that e-governance would play a moderating role in the relationship between innovation and e-service delivery. The findings ($\beta = 0.079$, $p < 0.01$) provided strong support for our hypothesis H6 as an indirect moderating relationship between innovation and e-service delivery. The outcomes showed a substantial and progressive direct and indirect relationship between e-governance and e-service delivery; e-governance plays a critical positive and significant moderating role between innovation and e-service delivery.

It is important to understand the essence of the variables, so we have explained the essence of the variables and their indicators in Table 5. In Model 1, the control variables were included; Model 2 explains independent variables, which are AI-based KM and innovation; Model 3 describes the moderating variable of e-governance; Model 4 indicates our moderating test of e-governance between the relationship of innovation and e-service delivery; and finally, Model 5 explains the mediating role of innovation between AI-based KM and e-service delivery. All of the six hypotheses that were analyzed in the five models were substantially supported. Further, a summary of the results regarding the development of technology-oriented knowledge management and its impact on e-service delivery, along with the mediating role of innovation and the moderating role of e-governance, are given in Table 6 below.

**Table 4.** Multiple regression analysis—the effect of AI-based KM, innovation, and e-governance on e-service delivery.

| Variables | Dependent Variable: E-Service Delivery | | | | DV: Innovation |
|---|---|---|---|---|---|
| | Model 1 | Model 2 | Model 3 | Model 4 | Model 5 |
| *Independent Variables* | | | | | |
| (Constant) | 4.500 (0.1118) *** | 1.207 (0.139) *** | 1.398 (0.144) *** | 2.309 (0.379) *** | 1.066 (0.080) *** |
| Education | 0.043 (0.017) *** | 0.005 (0.011) | 0.006 (0.011) | 0.004 (0.011) | 0.006 (0.007) |
| Gender | 0.099 (0.050) ** | 0.172 (0.033) *** | 0175 (0.032) *** | 0.177 (0.032) *** | 0.064 (0.021) *** |
| Exp | 0.300 (0.036) *** | 0.100 (0.024) *** | 0.091 (0.024) *** | 0.086 (0.024) *** | 0.026 (0.016) *** |
| Knowledge Mgt. | | 0.309 (0.053) *** | 0.388 (0.055) *** | 0.370 (0.055) *** | 0.766 (0.014) *** |
| Innovation | | 1.027 (0.063) *** | 1.283 (0.086) *** | 0.935 (0.159) *** | |
| E-Governance | | | 0.223 (0.051) *** | 0.447 (0.100) *** | |
| *Moderating effect* | | | | | |
| Innovation x E-Governance | | | | 0.079 (0.031) *** | |
| *Mediating effect* | | | | | |
| Knowledge Mgt. → Innovation → E-Service Delivery | | | | *(Sobel Test)* | 12.114 (0.282) *** |
| N | 569 | 569 | 569 | 569 | 569 |
| R | 0.344 | 0.798 | 0.805 | 0.808 | 0.931 |
| $R^2$ | 0.118 | 0.637 | 0.649 | 0.653 | 0.866 |
| Std. Error | 0.588 | 0.378 | 0.372 | 0.370 | 0.250 |
| F Models | 25.316 *** | 197.424 *** | 172.968 *** | 150.738 *** | 912.608 *** |
| Durbin-Watson | 1.704 | 1.993 | 2.032 | 2.045 | 1.882 |

** Correlation is significant at the 0.01 level (2-tailed); *** Correlation is significant at the 0.001 level (2-tailed).

**Table 5.** Characteristics of the evaluation models of artificial intelligence-based technological-oriented knowledge management, innovation, and e-service delivery in smart cities.

| | Essence | Indicators | Results (Effects) |
|---|---|---|---|
| Model 1 | Control Variables | Constants, Education, Gender, and Experience | Strongly Supported |
| Model 2 | Independent Variables | AI-based KM and Innovation | Strongly Supported |
| Model 3 | Moderating Variable | E-Governance | Strongly Supported |
| Model 4 | Moderating Test | Innovation x E-Governance | Strongly Supported |
| Model 5 | Mediating Test | AI-based KM, Innovation, and E-Service Delivery | Strongly Supported |

**Table 6.** Generalization of the hypotheses of artificial intelligence-based technological oriented knowledge management, innovation, and e-service delivery in smart cities.

| | Essence | Results (Effects) | Factors of Influence | |
|---|---|---|---|---|
| | | | Positive | Negative |
| Hypothesis 1 | AI-based technology-oriented knowledge management → E-service Delivery | Supported | Direct impact | - |
| Hypothesis 2 | AI-based technology-oriented knowledge management → Innovation success | Supported | Direct impact | - |
| Hypothesis 3 | Innovation → E-service Delivery | Supported | Direct impact | - |
| Hypothesis 4 | AI-based technology-oriented knowledge management → Innovation → E-service Delivery | Supported | Mediating impact | - |
| Hypothesis 5 | E-governance → E-service Delivery | Supported | Direct impact | - |
| Hypothesis 6 | Innovation → E-governance → E-service Delivery | Supported | Moderating impact | - |

## 5. Discussion

According to the knowledge management-based service science theory by [18] and the diffusion of innovation theory of [19], a city government should integrate its technological resources and competencies to manage acquired knowledge and enhance e-service delivery

through technological innovation. Following the theoretical framework, the findings corroborate our hypothesis that enhancing innovation must be driven by the interaction effects of knowledge management and city performance. Hess and Rothaermel [85] explored the role of innovation on a city's performance to determine when and how technology-oriented sources are substitutive. This paper advances a research gap by examining the interaction effects of technology-oriented knowledge management and e-service delivery on innovation and the contextual role of e-governance between innovation and e-service delivery. City governments must implement diverse approaches regarding e-service offering and e-service delivery protocols by ensuring innovation and e-governance, fostering good e-governance with innovation.

Table 4 shows a diverse range of results. All the correlations between knowledge management, innovation, e-governance, and smart city performance indicators were positive and statistically significant. Table 4 provides evidence that several independent knowledge management elements do not correlate with different aspects of performance measures. One plausible interpretation of these findings is that comparative performance metrics may struggle from a halo effect, wherein city governors sensationalize their own cities' effectiveness. Besides that, knowledge management is not the only factor that influences performance, and other factors, such as the city's innovative or e-government environment, may substantially impact performance. The relationship between knowledge management and innovation was theoretically established in the literature, but statistical evidence was inadequate.

Consequently, in this study, a city that is proficient in knowledge management attributes is more innovative. According to a common assumption, intangible knowledge is more complicated for contenders to access and replicate. Therefore, this type of knowledge has a tremendous opportunity to transform competitive advantages [86], improving performance. The findings that are presented in this study are significant because they demonstrate that knowledge is just as essential as what we do with that knowledge to be innovative.

Smart cities with well-developed technology-oriented knowledge management behavioral patterns are more likely to generate greater performance (i.e., e-service delivery) and develop incremental innovations supporting our proposed H1 and H2 substantially. Moreover, municipalities with well-developed innovations and technology are more strongly predictive of e-service delivery, with the fact that technological innovation is critical for providing electronic services in smart cities, supporting our assertion in H3. These conclusions are also supported by an analysis of individual knowledge management factors. Our empirical analysis not only suggests that knowledge management has a significant and positive influence on innovation and innovation had a significant positive effect on smart city performance, but the findings also revealed that knowledge management has a significant indirect effect on smart city performance through innovation, supporting our projected H4 substantially, suggesting that cities with more information technology can enhance performance by maximizing the e-services that they provide to their citizens.

Furthermore, our statistical analysis recommended that e-governance substantially and positively impacts smart city performance; therefore, our proposed H5 was supported significantly. The findings also supported H6 and proved that the e-governance factor strengthens the direct relationship between innovation and performance; hence this moderating relationship is also confirmed. In the context of smart cities through innovation, we investigated the role of e-governance in boosting e-service delivery and its implications on citizen satisfaction. According to the study findings, e-governance has the potential to strengthen the association between innovation and e-service delivery. There is a significant disparity in the expectations and perceptions of ordinary citizens in the cities regarding service delivery, which has harmed residents' satisfaction over the years. Considering the overall adverse effect of the predominant dilemma, there is an imperative need in developing cities that lack innovation to implement e-governance in all public agencies [87].

Further direction, strategic purposes, and measures to implement these strategic purposes are explained in Table 7.

**Table 7.** Directions and means of development of technology-oriented knowledge management based on artificial intelligence, innovations, and the provision of electronic services in smart cities.

| Parameters (Directions) | Strategic Purposes | Means (Measures) of Implementation of Strategic Purposes |
|---|---|---|
| Artificial Intelligence-based Technological-Oriented Knowledge Management | ○ Disaster management [88] <br> ○ Technological innovation and revolution [89,90] <br> ○ Firm growth and performance [91] <br> ○ Enhance business process [92] | ○ Strategic planning, mitigation and preparedness activities, rehabilitation <br> ○ Knowledge sharing, application and storage, learning and decision-making <br> ○ Competitive advantage <br> ○ Leadership support, adequate funds, functional support |
| Innovation | ○ Service innovation [52] <br> ○ E-services innovation [54,93] <br> ○ Technological innovation [89] | ○ Incorporation of product innovation and introduction of new products/services <br> ○ User interaction with products/services <br> ○ User/customer experience <br> ○ Acquisition of knowledge, software, and hardware to develop new services <br> ○ Investment in ICT |
| E-Governance | ○ Provision of E-services [94] <br> ○ Public development [95] | ○ Incorporation of private and non-profit IT projects <br> ○ E-administration <br> ○ Transparency between public-private businesses |
| E-Service Delivery | ○ Healthcare [96] <br> ○ Education [97] <br> ○ Social services [98] <br> ○ Other E-services [41] | ○ Decision-making about public healthcare <br> ○ Culture of education and provide practical tools to adapt management process <br> ○ Renew registration for a motor vehicle online <br> ○ Renew a driver's license online <br> ○ Renew an international passport online <br> ○ obtain a copy of a birth/marriage certificate for self electronically |
| Smart Cities (considering the total impact according to the selected parameters) | ○ Smart economy <br> ○ Smart governance <br> ○ Smart environment [99] <br> ○ Smart mobility <br> ○ Smart living | ○ Innovative business approach, R&D expenditures, labor market productivity, and city's economic role in the national/international market <br> ○ Use of ICT and participation of people in decision-making process <br> ○ Responsible resource management and sustainable urban planning <br> ○ Efficient transportation system <br> ○ Citizen's quality of life |

## 6. Conclusions

This study demonstrates that direct and indirect driving forces are mutually advantageous. Furthermore, analyzing their interaction can help to model the relationships between knowledge management, innovation, e-governance, and e-service delivery. Smart cities should manage the knowledge that is acquired through artificial intelligence and develop new information technology-based e-services through innovation. Furthermore, innovation mediates the relationship between knowledge management and e-service delivery, while e-governance moderates the relationship between innovation and smart city performance.

### 6.1. Theoretical Implications

Organizations make decisions about what operations the organization will engage in, how those operations will be carried out, what resources will be necessary, which

resources will be disbursed to different functions, and, eventually, which resources will be used [100]. In this context, this study contends that knowledge that is acquired through artificial intelligence serves several functions:

1. Technological-oriented knowledge can be both an intangible and tangible resource [101] that can be used for better decision-making.
2. Acquiring knowledge favors any decision-making regarding utilizing resources to provide electronic services.
3. A competency in knowledge management empowers everyone within a city government to capitalize the most assistance from the knowledge and other capabilities [100].
4. Effective, efficient, and constructive knowledge management contributes significantly to innovation.
5. Innovation through KM has a stronger influence over e-service delivery when a smart city has a high degree of e-governance.

Constructive knowledge management was developed as a coordinating mechanism by presenting substantial evidence with a proclivity for establishing innovation capabilities were more likely to have well-developed knowledge management policies and attitudes. It is reasonable to suggest that most smart cities have knowledge management capabilities and ensure the effective utilization of other accessible resources. This finding provides early evidence for [100] concepts by demonstrating the importance of knowledge management as a coordinating mechanism when formulating innovation capabilities. Furthermore, we discovered substantial evidence for the notion that a smart city that was developing dynamic innovations had well-developed knowledge management policies and behaviors, as well as credible evidence that enhanced smart city performance and knowledge management co-existed.

Technology-oriented knowledge management was found to directly impact e-service delivery and innovation, while innovation had a direct effect on e-service delivery. When e-governance was added as a moderator, it not only directly impacted e-service delivery but also strengthened the relationship between innovation and e-service delivery. These findings are significant because empirical support is provided for the existing knowledge management-based service science theory [18] and the diffusion of innovation theory of [19], and, more importantly, empirically evidenced development of e-governance as a moderator between the innovation and e-service delivery is yet another contribution to the literature of innovation and applied sciences.

*6.2. Managerial Implications*

Knowledge management has been heralded as a novel discipline. The understanding of the concept of knowledge management is frequently systematic with the advent of information technology as a remedy for knowledge acquisition. This study addresses a broader knowledge management framework by utilizing previously discovered knowledge management elements that are characteristics of an organization that manages knowledge effectively [76]. The study also demonstrates the significance of effective knowledge management. Consequently, smart city managers should develop initiatives to improve knowledge management attitudes and behaviors because a city that manages knowledge effectively will be more innovative. Furthermore, smart city governors should develop and implement an e-governance system to improve e-service delivery to smart city citizens through innovative technologies.

*6.3. Limitations and Future Research*

Similar to most empirical research, this study has certain limitations that must be considered when interpreting, extending, and generalizing the findings. Since this research was performed in Asian countries such as South Korea, Pakistan, and Japan, the attributes of the analyzed respondents may not extend to those in other cultures and countries that differ from those that were mentioned. Consequently, further investigation into cross-continent differences in social mechanisms that are designed to address innovation

in e-service delivery is warranted. Finally, because participation in this survey was discretionary, consciousness variance was unavoidable. The Harman one-factor test was used to rule out any potential issues. According to the results of the test, each major construct describes roughly equal variance, denoting that our data do not have an elevated common method variance.

According to the findings of this study, smart cities that effectively manage knowledge were more innovative and outperformed in delivering e-services. The study also discovered that knowledge management influenced innovation and that innovation influenced performance positively, and e-governance significantly impacted performance and moderated the relationship between innovation and performance. One of the core themes of this study is that effective knowledge management facilitates the extraction of high-quality e-services from certain resources. Future research is needed to strengthen and expand this assumption by investigating the facilitating importance of knowledge management in greater depth.

**Author Contributions:** S.A.A.B., conceptualization, methodology, validation, formal analysis, investigation, and writing—original draft preparation; S.M., conceptualization, methodology, writing—review and editing, and funding acquisition. All authors have read and agreed to the published version of the manuscript.

**Funding:** This work was supported by the Ministry of Education of the Republic of Korea and the National Research Foundation of Korea (NRF-2019S1A5C2A03081234).

**Institutional Review Board Statement:** Not applicable.

**Informed Consent Statement:** Not applicable.

**Data Availability Statement:** Not applicable.

**Conflicts of Interest:** The authors declare no conflict of interest.

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
