# Peer review of "Artificial Intelligence-Based Technological-Oriented Knowledge Management, Innovation, and E-Service Delivery in Smart Cities: Moderating Role of E-Governance"

_applsci, doi:10.3390/app12178732_

Round 1
Reviewer 1 Report
In the article, it is necessary to summarize the individual results of the research, and especially the economic and mathematical calculations, and to more thoroughly characterize the strategic priorities of the development of artificial intelligence-based technologically-oriented knowledge management, innovation, and e-service delivery in smart cities.
Thank you!

Author Response
Dear Reviewer,
We appreciate the kind review of our manuscript, valuable comments, and thoughtful suggestions. We have included three tables with details, which are given below. We also have added them to the manuscript in their appropriate places to improve the quality of the paper. We hope our paper will add value to the research of AI-based KM and its impact on service delivery in smart cities. Responses to valuable comments are as under:
Globalization changes confirm the increasing spread of digital technologies in the social and economic life of the population. Of great importance are technologies that ensure the improvement of knowledge, the introduction of new viability methods, and the improvement of the quality of state regulation. The article's authors paid attention to such aspects of the country's socio-economic development.
Despite the progressiveness of the research direction, the authors must take into account the following debatable points:
1) it is appropriate to summarize the authors' assumptions regarding developing technology-oriented knowledge management and build a table.
Response: A summary of the results regarding the development of technology-oriented knowledge management and its impact on e-service delivery, along with mediating role of innovation and moderating role of e-governance, are given in Table 5 in the manuscript on page 13, please.
Table 6: Generalization of the hypotheses of artificial intelligence-based technological-oriented knowledge management, innovation, and e-service delivery in smart cities
|
Essence |
Results (Effects) |
Factors of Influence |
|
Positive |
Negative |
|||
Hypothesis 1 |
AI-based technology-oriented knowledge management à E-service Delivery |
Supported |
Direct impact |
- |
Hypothesis 2 |
AI-based technology-oriented knowledge management à Innovation success |
Supported |
Direct impact |
- |
Hypothesis 3 |
Innovation à E-service Delivery |
Supported |
Direct impact |
- |
Hypothesis 4 |
AI-based technology-oriented knowledge management à Innovation à E-service Delivery |
Supported |
mediating impact |
- |
Hypothesis 5 |
E-governance à E-service Delivery |
Supported |
Direct impact |
- |
Hypothesis 6 |
Innovation àE-governance à E-service Delivery |
Supported |
moderating impact |
- |
2) in points 3.2.1, 3.2.2, 3.2.3, and 3.2.4, it is appropriate to indicate the abbreviated names of indicators as in table 2, which gives an accurate understanding of their essence.
Response: We appreciate the valuable comment. We have inserted abbreviated names of indicators in the appropriate places in 3.2.1, 3.2.2, 3.2.3, and 3.2.4 on pages 8 and 9.
3) the essence of the models 1-4 proposed in the calculations (pages 11-13), which indicate the control constructors (what are the indicators), the main variables (what are these variables), and other statements, is not clear. It is appropriate to clarify the essence of models (1-4)
Response: Please add details in the manuscript along with table 5 on page 13.
Table 5: Characteristics of evaluation models of artificial intelligence-based technologicaloriented knowledge management, innovation, and e-service delivery in smart cities
|
Essence |
Indicators |
Results (effects) |
Model 1 |
Control Variables |
Constants, Education, Gender, and Experience |
Strongly Supported |
Model 2 |
Independent Variables |
AI-based KM and Innovation |
Strongly Supported |
Model 3 |
Moderating Variable |
E-Governance |
Strongly Supported |
Model 4 |
Moderating Test |
Innovation x E-Governance |
Strongly Supported |
Model 5 |
Mediating Test |
AI-based KM, Innovation, and E- Service Delivery |
Strongly Supported |
4) in point 5 of the «discussion,» it is appropriate to highlight the prospects for developing technology-oriented knowledge management based on artificial intelligence, innovations, and the provision of electronic services in smart cities and to build the following table.
Response: Please add details in the manuscript along with table 6 on page 15.
Table ** Directions and means of development of technology-oriented knowledge management based on artificial intelligence, innovations and the provision of electronic services in smart cities
Parameters (directions) |
Strategic purposes |
Means (measures) of implementation of strategic purposes |
Artificial Intelligence-based Technological-Oriented Knowledge Management |
§ Disaster management (Oktari et al., 2020) § Technological innovation and revolution (Lee et al., 2013; Russ, 2021) § Firm growth and performance (Hitt et al., 2000) § Enhance business process (Chatterjee et al., 2020) |
§ Strategic planning, mitigation and preparedness activities, Rehabilitation § Knowledge sharing, application and storage; learning and decision-making § Competitive advantage § Leadership support, adequate funds, functional support |
Innovation |
§ Service innovation (Oke, 2007) § E-services innovation (Ciuchita et al., 2019; Tsou & Chen, 2012) § Technological innovation (Lee et al., 2013) |
§ Incorporation of product innovation and introduction of new products/services § User interaction with products/services § User/customer experience § Acquisition of knowledge, software, and hardware to develop new services |
E-Governance |
§ Provision of E-services (Bannister & Connolly, 2012) § Public development (Madon, 2009) |
§ Investment in ICT § Incorporation of private and non-profit IT projects § E-administration § Transparency between public-private businesses |
E-Service Delivery |
§ Healthcare (El Morr & Subercaze, 2010) § Education (Sallis & Jones, 2013) § Social services (Gabbay et al., 2003) § Other E-services (Erastus et al., 2021) |
§ Decision-making about public healthcare § Culture of education and provide practical tools to adapt management process § Renew registration for a motor vehicle online § Renew a driver’s license online § Renew an international passport online § obtain a copy of a birth/marriage certificate for self electronically |
Smart Cities (taking into account the total impact according to the selected parameters) |
§ Smart economy § Smart governance § Smart environment (Colldahl et al., 2013) § Smart mobility § Smart living |
§ Innovative business approach, R&D expenditures, labor market productivity, and city’s economic role in the national/international market § Use of ICT and participation of people in decision-making process § Responsible resource management and sustainable urban planning § Efficient transportation system § Citizen’s quality of life |

Reviewer 2 Report
The fundamental goal of this research is to investigate the quantitative relationship between technology-oriented KM, innovation, e-governance, and smart city performance using KM-based service science theory and diffusion of innovation theory. The overall quality of the article is in line with the expectation. However, the authors may consider the following comments in the revised version.
Introduction
Generally, the introduction should first present the nature and scope of the problem investigated. The paper proposed in the first paragraph, "The appraisal of the KM infrastructure that allows the institutions to identify, develop, transform, and disseminate knowledge is crucial to understanding the strengths and weaknesses of KM initiatives." I think it's the main aim, not the key problem in this paper, so I suggest making it more relevant to the key problem.
It should state the reasons for choosing a particular method. For objectiveness, why did you choose this method?
In lines 83-85, the author proposed the study's main objective. However, I think this sentence is hard to follow.
Methodology
The data was acquired from a sample of South Korea, Pakistan, Bangladesh, and Japan. I suggest describing why you choose these areas.
Result
The captions of the figures and tables are too simple; they should explain the image and the figure's content in detail.
The data are captured from different countries and areas; therefore, I suggest stating their differences.
Discussion
I think the significance of the results is not discussed adequately. The fundamental goal of this research is to investigate the quantitative relationship between technology-oriented knowledge management... To emphasize this point, the Discussion need to show the relationships among observed facts.
Author Response
Dear Reviewer,
We appreciate your kind review of our manuscript, valuable comments, and thoughtful suggestions. We have included three tables with details as the Reviewer suggested, which are given below. We also have added them to the manuscript in their appropriate places to improve the quality of the paper. We hope our paper will add value to the research of AI-based KM and its impact on service delivery in smart cities. Responses to the valuable comments are as under:
The fundamental goal of this research is to investigate the quantitative relationship between technology-oriented KM, innovation, e-governance, and smart city performance using KM-based service science theory and diffusion of innovation theory. The overall quality of the article is in line with the expectation. However, the authors may consider the following comments in the revised version.
- Introduction
Generally, the introduction should first present the nature and scope of the problem investigated. The paper proposed in the first paragraph, "The appraisal of the KM infrastructure that allows the institutions to identify, develop, transform, and disseminate knowledge is crucial to understanding the strengths and weaknesses of KM initiatives." I think it's the main aim, not the key problem in this paper, so I suggest making it more relevant to the key problem.
It should state the reasons for choosing a particular method. For objectiveness, why did you choose this method? In lines 83-85, the author proposed the study's main objective. However, I think this sentence is hard to follow.
Response:
We appreciate the valuable comment, and we tried to respond to the comment accordingly at our best.
Introductions are challenging to write as it is important to get to the paper's point quickly and with sufficient motivation. In our paper, the main objective is to examine the following relationships empirically:
- Impact of AI-based KM, innovation, and e-governance on e-service delivery
- The mediating role of innovation
- The moderating role of e-governance
These are the research gaps that we found in the previous literature. To write an introduction, we followed 4 steps rules which are:
- What is known about the topic is given in paragraphs one to three.
- What is unknown about the topic is given in paragraph four.
- Our research question/main objective of the study is in paragraph five.
- We are given what we did and its implications at the end of paragraph five.
The mentioned sentence, "The appraisal of the KM infrastructure that allows the institutions to identify, develop, transform, and disseminate knowledge is crucial to understanding the strengths and weaknesses of KM initiatives," we modified according to our research requirement.
- Methodology
The data was acquired from a sample of South Korea, Pakistan, Bangladesh, and Japan. I suggest describing why you choose these areas.
Response:
Thank you very much for your kind and valuable comment. Here is the justification.
The justification for choosing these four countries was that South Korea and Japan are East Asian developed economies with strong e-governance and e-services for their citizens (Park et al., 2009). In contrast, Pakistan and Bangladesh are South Asian emerging economies striving to design and implement such governance and services (Jamil, 2021), so it is essential to evaluate the respondents' perceptions from different geographic areas from the same continent. According to our best knowledge and observation, only a few studies have yet been undertaken in the comparative sense of such Asian regions (Chaudhary et al., 2016).
Please provide this detail on page 7 from lines number 336-343.
- Result
The captions of the figures and tables are too simple; they should explain the image and the figure's content in detail.
Response:
Thank you very much for your very valuable comment. Captions of the figures and tables are provided in detail now, and they can be checked in tracking on Line number 355, 454, 477, 490, 501, and 516, please.
- Discussion
I think the significance of the results is not discussed adequately. The fundamental goal of this research is to investigate the quantitative relationship between technology-oriented knowledge management... To emphasize this point, the Discussion needs to show the relationships among observed facts.
Response:
Thank you very much for your very valuable comment. As the Reviewer suggested, Table 7 is added to the Discussion to explain the relationship between AI-based KM, innovation, e-governance, and e-service delivery, along with their parameters, strategic purposes, and means to implement those strategic purposes on pages 15 and 16.

Round 2
Reviewer 1 Report
Thanks to the authors for the responsible consideration of the recommendations, which allows recommending the article for publication in the journal.
Reviewer 2 Report
The authors have addressed my concerns.